# Genomic biosurveillance detects a sexual hybrid in the sudden oak death pathogen

Richard C. Hamelin [1✉], Guillaume J. Bilodeau [2], Renate Heinzelmann[1,3], Kelly Hrywkiw[1], Arnaud Capron[1], Erika Dort[1], Angela L. Dale[4], Emilie Giroux[2], Stacey Kus[4], Nick C. Carleson[5], Niklaus J. Grünwald [5,6] & Nicolas Feau [1✉]

Invasive exotic pathogens pose a threat to trees and forest ecosystems worldwide, hampering the provision of essential ecosystem services such as carbon sequestration and water purification. Hybridization is a major evolutionary force that can drive the emergence of pathogens. *Phytophthora ramorum*, an emergent pathogen that causes the sudden oak and larch death, spreads as reproductively isolated divergent clonal lineages. We use a genomic biosurveillance approach by sequencing genomes of *P. ramorum* from survey and inspection samples and report the discovery of variants of *P. ramorum* that are the result of hybridization via sexual recombination between North American and European lineages. We show that these hybrids are viable, can infect a host and produce spores for long-term survival and propagation. Genome sequencing revealed genotypic combinations at 54,515 single nucleotide polymorphism loci not present in parental lineages. More than 6,000 of those genotypes are predicted to have a functional impact in genes associated with host infection, including effectors, carbohydrate-active enzymes and proteases. We also observed post-meiotic mitotic recombination that could generate additional genotypic and phenotypic variation and contribute to homoploid hybrid speciation. Our study highlights the importance of plant pathogen biosurveillance to detect variants, including hybrids, and inform management and control.

[1] The Department of Forest and Conservation Sciences, University of British Columbia, Vancouver, BC, Canada. [2] Ottawa Plant Laboratory, Canadian Food Inspection Agency, Ottawa, ON, Canada. [3] Swiss Federal Institute for Forest, Snow and Landscape Research WSL, Birmensdorf, Switzerland. [4] New Construction Materials, FPInnovations, Vancouver, BC, Canada. [5] Department of Botany and Plant Pathology, Oregon State University, Corvallis, OR, USA. [6] Horticultural Crops Research Unit, USDA ARS, Corvallis, OR, USA. ✉email: richard.hamelin@ubc.ca; nicolas.feau@NRCan-RNCan.gc.ca

Emerging infectious diseases caused by filamentous pathogens pose a threat to animals, crops and forest ecosystems worldwide[1]. Globalization is driving the spread of pathogens into naïve host populations and biotic homogenization provides an advantage to generalists[2]. Disease outbreaks can have devastating ecological, economic and social impacts; they can reduce biodiversity, displace native species and threaten food security[2]. Pests and pathogens can affect landscape resilience and reduce the ability of forests to provide essential ecosystem services such as carbon sequestration and water purification[3].

Hybridization can favor the emergence of infectious diseases by generating novel pathogen lineages with increased fitness that can colonize new niches[4–9]. The large-scale genomic changes that result from hybridization generate new genotypic and phenotypic combinations upon which natural selection can act to modify traits such as pathogenicity and transmission[10,11]. Selective forces can create the conditions that benefit the emergence of lineages with distinct traits that can result in lifestyle or host shifts. For example, human pathogens such as opportunistic yeasts and the malaria parasite have emerged following hybridization[6,12]. Plant pathogens have also emerged following hybridization events that facilitated host jumps[9], increased virulence[13,14] and host range expansion[15].

*Phytophthora ramorum* (Oomycota) is an exotic plant pathogen that emerged in the 1990s and is regulated and targeted for eradication and quarantines in Europe and North America. It can attack over 125 plant species and is responsible for the sudden oak death in California and Oregon[16] and sudden larch death in the UK[17] and France[18]. The pathogen comprises clonal lineages[19] that are reproductively isolated and diverged between 0.75 and 1.25 million years ago (MYA)[19,20]. Lineages NA1 and NA2 are confined to North America and lineage EU2 is only found in western Europe. Lineage EU1 is broadly distributed in Europe and the West Coast of North America[21,22]. Eight additional lineages were recently described in Asia[23]. Recombination can only take place between individuals of opposite mating types and was demonstrated in the laboratory between lineages EU1 (mating type A1) and NA1 (mating type A2) but the progeny displayed aberrant genotypic and phenotypic variation[24–26].

Preventing the emergence of novel pathogens requires the timely acquisition of data that can inform mitigation actions. Increasingly, pathogen genome sequencing is used within a biosurveillance framework to generate population genomic data across geographic and temporal scales that can be used to identify variants, recombinants and hybrids and inform pathway and risk analyses[27]. Because of the potential for *P. ramorum* inter-lineage hybridization in western North America where lineages of opposite mating types co-occur[22], we analyzed the genomes of

samples from Canada, the USA and Europe to characterize the patterns of variation and identified hybrids among two clonal lineages of the pathogen.

## Results

**Hybridization between divergent clonal lineages**. We discovered hybrids between the EU1 (mating type A1) and NA2 (mating type A2) clonal lineages of *P. ramorum* from a single nursery in British Columbia, Canada, where these lineages had been previously reported. The qPCR lineage genotyping pattern of isolates 16-237-021 and 16-284-032, collected from infected rhododendron plants, do not match the pattern of the known clonal *P. ramorum* lineages in North America and Europe (Supplementary Table 1). We analyzed the genomes of 95 *P. ramorum* isolates (Supplementary Table 2) from Europe and North America and mapped the sequence reads onto the reference genome, yielding 450,656 single nucleotide polymorphisms (SNPs). Of the 31,047 SNPs that were homozygous for different alleles in the EU1 and NA2 lineages, 96.6% were heterozygous in isolates 16-237-021 and 16-284-032 (Table 1). Genome-wide heterozygosity was nearly as high in the hybrid ($H_o = 34.59\%$) as in the EU1 lineage ($H_o = 35.22\%$; Supplementary Table 3). All other lineages, including the other putative parent of the hybrid, had lower heterozygosity (Table 1). A principal component analysis of all 450,656 SNPs placed the two putative hybrids between the EU1 and NA2 lineages and ancestry estimation assigned them with equal probability to those putative parental lineages (Fig. 1a, b). A phylogenetic network analysis placed the hybrids in a branch with shared reticulations with lineages EU1 and NA2, the pattern expected for recombination (Fig. 1c).

**First-generation homoploid hybrids generated via sexual recombination**. The hybrids are likely homoploid (without a change in chromosome copy number; Supplementary Fig. 1) and result from a first generation (F1) recombination event. Analysis of phased genomic regions revealed distinct haplotypes at nuclear loci in the hybrids clustering either with a NA2 or an EU1 haplotype (Fig. 2a–c). The contribution of the putative parents to the nuclear genome of the hybrids was equal (Z-score = 8.78 ± 0.24; $p < 0.0001$; Supplementary Table 4) and they were assigned to a simulated EU1 x NA2 population with ≥0.999 probability (assignment to all other populations <0.0001; Supplementary Fig. 2a). A neighbor-joining tree clustered the hybrid samples with simulated first-generation hybrids but not with simulated populations backcrossed to either parental lineage (Supplementary Fig. 3). The two hybrid samples share genotypes at 99.8% of the SNP loci and are thus likely clones derived from a single hybridization event, via sexual

**Table 1 Genotypic combinations in *Phytophthora ramorum* parental lineages and hybrid.**

| Parental lineage genotypes | Hybrid genotype | Observed genotypes | Expected genotypes (sexual recombination) | Expected genotypes (heterokaryosis) |
|---|---|---|---|---|
| 0/0 × 1/1 | 0/0 or 1/1 | 1,049 (3.4%) | 0 (0.0%)[b] | 0 (0.0%)[b] |
|  | 0/1[a] | 29,998 (96.6%) | 31,047 (100.0%)[b] | 31,047 (100.0%)[b] |
| 0/1 × 0/1 | 0/0[a] | 12,295 (24.7%) | 12,465 (25.0%)[b] | 0 (0%)[c] |
|  | 1/1[a] | 12,222 (24.5%) | 12,465 (25.0%)[b] | 0 (0%)[c] |
|  | 0/1 | 25,343 (50.8%) | 24,930 (50.0%)[b] | 49,860 (100%)[c] |
| 0/1 x (0/0 or 1/1) | 0/0 or 1/1 | 114,604 (50.1%) | 114,510 (50.0%)[b] | 0 (0%)[c] |
|  | 0/1 | 114,415 (49.9%) | 114,510 (50.0%)[b] | 49,860 (100%)[c] |

Observed percentage of SNPs in each genotypic combination in the parental lineages and the *P. ramorum* hybrid samples 16-237-021 and 16-284-032 and expected genotypic combinations with sexual recombination and heterokaryosis following hyphal fusion (anastomosis).
[a]Genotypic combinations in the hybrid not observed in the parental lineages, totaling 54,515 SNPs.
[b]Expected genotype frequencies are not different from expected under the recombination model (chi-square test performed on 1000 subsamples of 100 random loci; $p > 0.05$ for ≥ 95.0% of the subsamples).
[c]Observed genotype frequencies are different from expected under the recombination model (chi-square test performed on 1000 subsamples of 100 random loci; $p \leq 0.05$ for 100% of the subsamples).

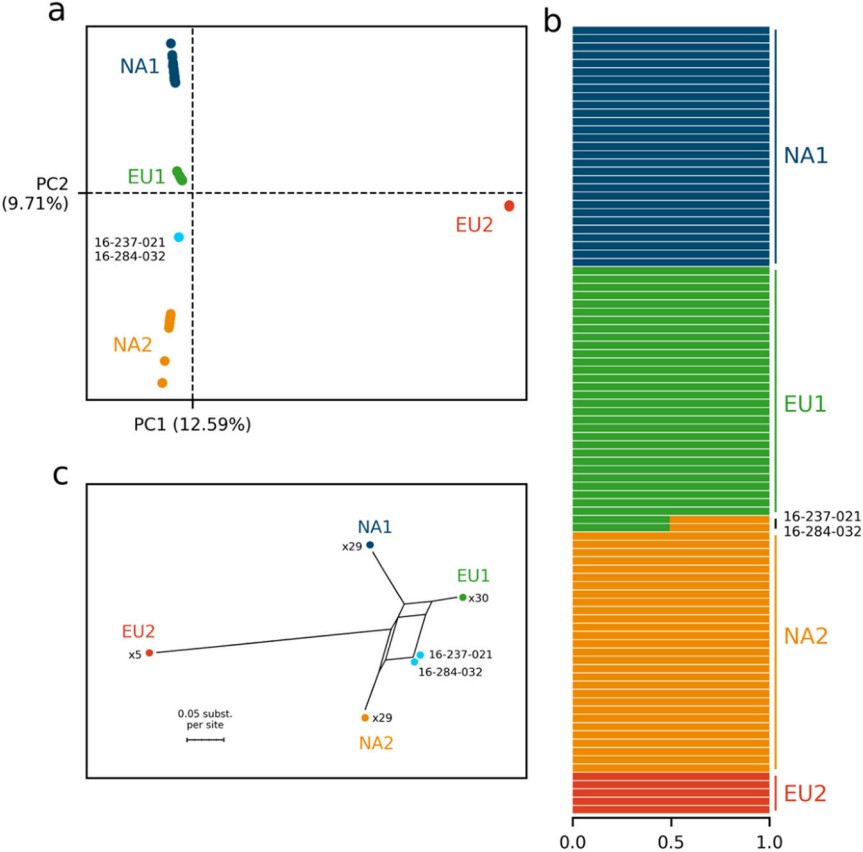

**Fig. 1 Hybridization between divergent clonal lineages EU1 and NA2 of *Phytophthora ramorum*.** We analyzed 95 whole genomes of *P. ramorum* and extracted 450,656 single nucleotide polymorphisms (SNP) to characterize the populations; **a** Population structure analysis of *P. ramorum* using a principal component analysis; each dot represents the genome of a *P. ramorum* isolate; **b** Ancestry estimation using Admixture analysis of *P. ramorum* lineages and putative hybrids at K = 4; each bar represents the genome of a *P. ramorum* isolate; **c** Neighbor-net phylogenetic network reconstructed from a matrix of pairwise Nei's genetic distances between isolates of *P. ramorum*. Samples 16-237-021 and 16-284-032 are the two putative hybrids.

recombination, followed by clonal propagation. The observed genotypic combinations in the hybrids and the parental lineages support the hypothesis of sexual recombination (except for some excess of homozygosity observed in the hybrids [Table 1; see below]) but not of heterokaryosis following hyphal fusion (anastomosis; Table 1). The NA2 lineage likely acted as the "female" parent. We mapped the sequence reads of all 95 sequenced genomes to the mitochondrial genome of *P. ramorum* and retrieved their mitochondrial haplotypes (mitotypes). There are 103 polymorphic sites in the mitochondrial genomes of the four lineages of *P. ramorum*. The mitotypes of the EU1 and NA2 lineages differ at 42 positions. The NA2 and hybrid *P. ramorum* mitotypes are identical (with the exception of two SNPs in sample 16_0284_0032; Fig. 2d). This pattern indicates uniparental transmission of the mitochondrial genome[24].

**Predicted functional impact of hybridization.** The hybridization between two *P. ramorum* lineages that diverged approximately 1 MYA[19,20] could impact fitness. These lineages differ in several traits, including growth, sporulation and aggressiveness during host infection[23,28]. The growth rate of the hybrid isolates was intermediate between the parental lineages but significantly higher than the EU1 parent ($p = 3.75\text{e-}5 \times 10^{-5}$; Fig. 3d). The hybrids can produce both chlamydospores and sporangia (Fig. 3a, b, c), spores that are important for survival and spread. The hybrids were infectious on rhododendron, a common host, and caused lesions with sizes that overlapped with those of the other lineages (Fig. 3e).

The hybrids have genotypes not previously observed at 54,515 SNPs that are either homozygous for different alleles in the parental lineages but heterozygous in the hybrid, or heterozygous in both parental lineages but homozygous in the hybrid (Table 1). Of those SNPs, 6,736 are non-synonymous mutations and 6,752 are predicted to have a moderate (e.g., non-synonymous mutations) to high (e.g., stop codons, frameshifts) impact (Supplementary Tables 5 and 6). Several of the genes that have novel genotypic combinations in the hybrids are associated with pathogenicity[29–31], including 51 RxLR and crinkler-like effectors[31], 79 carbohydrate active enzymes (CAZymes), and 11 peptidases (Supplementary Table 7).

**Mitotic recombination generates additional variation.** Mitotic recombination in the hybrid could provide additional genotypic and phenotypic variation in the pathogen population. We found 1,049 SNPs that were homozygous for different alleles in the parental lineages and also homozygous for one of those alleles in the hybrids, an unexpected pattern for sexual recombinants (Table 1). We observed that 90% of those SNPs were distributed in stretches of 2 to 82 (median = 4.0) contiguous homozygous positions (runs of homozygosity; ROH) in the hybrid (Supplementary Fig. 4), indicating that mitotic recombination had occurred in these regions. These ROH were in gene-poor regions enriched in transposable elements where RxLR effectors and putative avirulence factors are present (Supplementary Table 8).

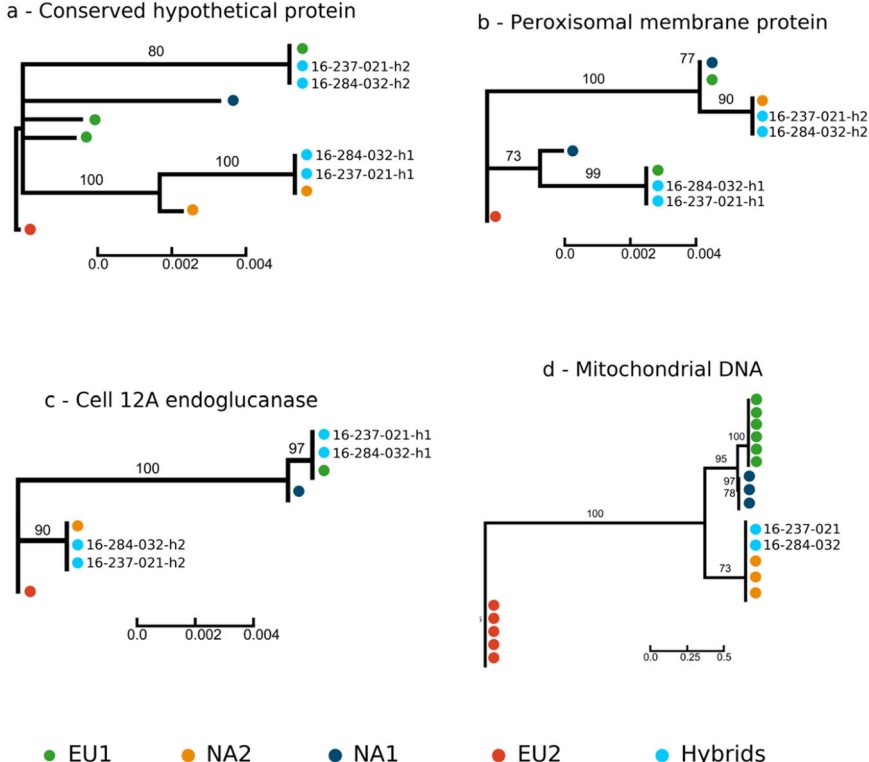

**Fig. 2 Haplotype phylogeny shows independent assortment of alleles at nuclear loci and uniparental inheritance of mitochondrial haplotype in the** *Phytophthora ramorum* **hybrid.** Neighbor-joining tree of haplotypes of *P. ramorum* indicates meiotic recombination with nuclear phased haplotypes clustering with haplotypes of one of the two parents and mitochondrial haplotypes clustering with lineage NA2. Phased haplotypes were obtained by using short genome regions with physical phasing when two or more variants co-occur on the same sequencing read. The following genomes were used: 14-1270, 15-1093 (EU1), 04-0372 (NA1), 5-0954 (NA2) and P2111 (EU2) for **a** Conserved hypothetical protein, **b** peroxisomal membrane protein and **c** cell 12 A endoglucanase. It is not unexpected to have more than two alleles at the conserved hypothetical protein for the EU1 lineage since two isolates were used. For the mitochondrial genomes **d**, the following samples were used: 03-0110, 05-7036, 08-3469, BBA-26-02, CC1048, P2599 (EU1); 04-0372, MK548, Pr1556 (NA1), 04-25165, 05-18753, 13-0781 (NA2); P2111, P2460, P2461 (EU2). The mitochondrial genome comprised 103 SNPs. For the two hybrid samples (16-237-021 and 16-284-032), the two haplotypes are indicated with h1 and h2.

## Discussion

The discovery of hybridization among lineages of *P. ramorum* should cause concerns in the plant and forest health communities. Hybridization provides a source of genetic variation[10] upon which natural selection can act to modify traits such as pathogenicity and transmission[11]. Selective forces could create the conditions for homoploid hybrid speciation[32] and the emergence of lineages with distinct traits. Neither hybridization nor recombination have previously been reported in the invasive range of *P. ramorum*. Previous studies that reported artificial crosses between *P. ramorum* EU1 and NA1 lineages produced progeny with reduced viability but that exhibited a broad range of pathogenicity[24–26]. This could be explained by transgressive trait variation, a common outcome of hybridization in plants, where hybrid phenotypes exceed those of the parents, contributing to ecological niche divergence[33].

We show that the EU1 and NA2 lineages can form stable and viable hybrids in nature that can sporulate and infect a plant host. The growth phenotypes we observed in the hybrids were intermediate compared to the parental phenotypes. However, we have only conducted inoculations on a single host, rhododendron. Our genome analyses indicate that hybridization has a potential functional impact, with thousands of amino acid changes, several predicted to have a large effect. Some of the genes affected are associated with host infection and include effectors, known to be involved in pathogen–host interactions[29–31]. A single amino acid polymorphism in a pathogen effector was recently shown to

expand its binding spectrum, allowing the pathogen to adapt to new hosts, and demonstrating the advantage that novel mutations can confer[34]. The genomic changes we report in the *P. ramorum* hybrids have the potential to impact diversification and adaptation.

The presence of sexual recombination among divergent *P. ramorum* lineages could modify the epidemiologic profile of the disease. Oospores, the spores produced following sexual reproduction, can survive adverse conditions such as drought and freeze and remain viable in the soil for years, making it possible for the pathogen to survive in the absence of a host[35] thereby increasing the risk of accidental propagation. In the potato famine pathogen, *Phytophthora infestans*, sexual recombination became possible when lineages of the opposite mating type migrated from the center of origin[36–39]. This increase in diversity contributed to the emergence of novel virulence combinations[40], coinciding with the appearance of lineages that were resistant to fungicides[39] and outbreaks that started earlier in the season, generating an increased disease risk[41].

In addition to meiotic recombination, we also observed mitotic recombination in the hybrids. Mitotic recombination was shown to accelerate adaptation in fungi[42] and appears to be an important mechanism fueling evolution in *P. ramorum*, producing ROH in all lineages[20]. Virulence differences and adaptation to environmental changes such as exposure to oxidative or heat stress and antifungal drugs have been associated with ROH in fungi[43] and could increase genotypic and

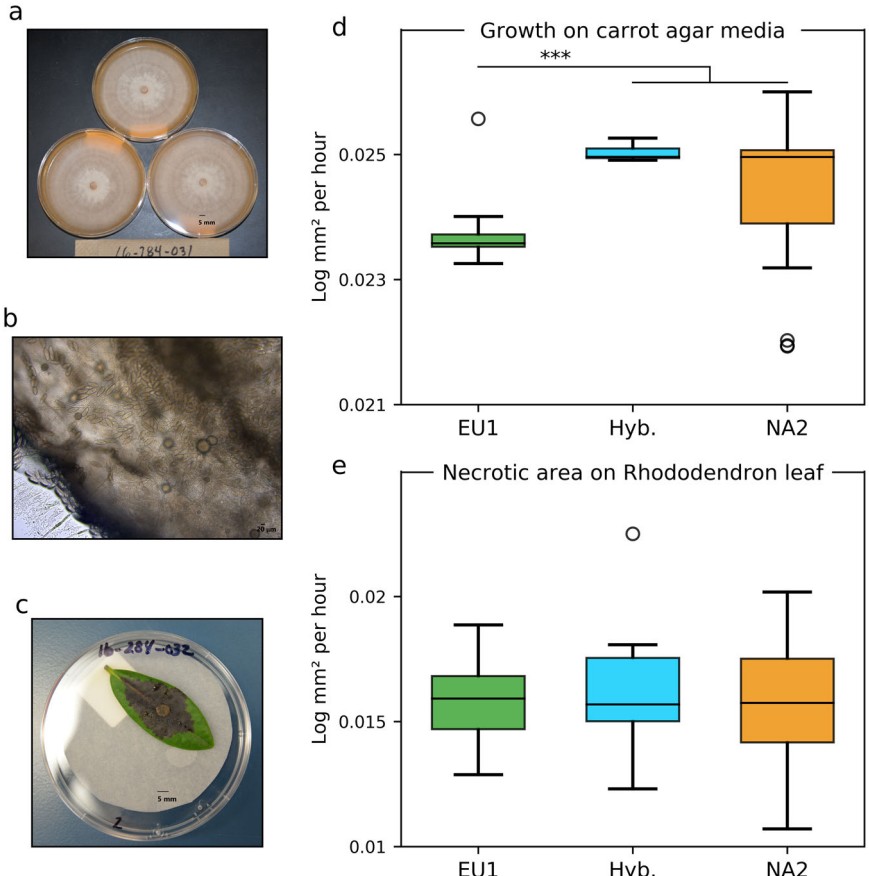

**Fig. 3 *Phytophthora ramorum* hybrid can sporulate, infect a host and has intermediate phenotypes. a** Morphology of *P. ramorum* hybrid isolate 16-284-032 growing on carrot agar, showing fluffy growth where sporangia are produced; **b** Sporangia produced by the *P. ramorum* hybrid in culture; **c** Rhododendron leaves infected by hybrid *P. ramorum* isolate 16-284-032; **d** Growth of *P. ramorum* lineages and hybrids on carrot agar medium; ***$p < 0.0001$, one-way ANOVA with multiple comparisons using Tukey test; **e** Growth of *P. ramorum* lineages and hybrids measured as necrotic area (π x lesion length x width) over time on rhododendron leaves; there was no significant difference in lesion growth among the lineages and hybrids. The lower and upper boundaries of each box in the boxplots (**d**, **e**) indicate limits of the interquantile range (IQR) between the 25th (Q1) and 75th percentile (Q3). Bars (whiskers) below and above the box indicate the minimum (Q1 – 1.5*IQR) and maximum (Q3 + 1.5*IQR) values of the distribution. The horizontal line in each box represents the median and outliers are represented with circles.

phenotypic diversity and divergence between the *P. ramorum* lineages and the hybrid.

So far, the *P. ramorum* EU1 × NA2 hybrid was only found in a single nursery in British Columbia, where the pathogen has not spread to natural forests. The disease was apparently eradicated in that nursery, preventing propagation of the hybrid. In Oregon, three lineages (NA1, NA2 and EU1) of *P. ramorum* have now spread to natural forests[44,45]. This generates the potential for hybridization among those lineages in forests; the spread of a hybrid lineage would be more difficult to contain in natural forests than in nurseries where containment and eradication are possible. Our study highlights the importance of genomic biosurveillance for the detection of plant pathogen variants and hybrids to inform mitigation strategies[27].

## Methods

**Sample collection and genomic characterization.** We obtained isolates of *P. ramorum* from nurseries in British Columbia (BC), Canada between 1995 and 2017 (Supplementary Table 2 and Supplementary Data 1). Leaf samples showing lesions were tested by qPCR for presence of *P. ramorum*[46] and cultures were obtained from positive samples by plating the lesions on culture media. Lineages of the *P. ramorum* isolates used for genome sequencing were determined with qPCR assays that targets four SNPs in the cellulose binding elicitor lectin (CBEL) locus using an Allele-Specific Oligonucleotide (ASO) assay[47]. These isolates were grown on 5% V8 juice agar[48] overlaid with a cellophane membrane. After 7 to 10 days, the mycelium was harvested and immediately frozen at −20 °C. The mycelium was shock frozen

in liquid nitrogen and pulverized with a Mixer Mill (Qiagen, Hillden, Germany) for 30 seconds at a frequency of 1/30.

DNA was extracted from approximately 200 mg of tissue following a chloroform extraction protocol including RNAase treatment[49]. DNA was quantified using the Qubit dsDNA BR assay Kit following manufacturer's instructions. Illumina libraries were prepared from 100 ng DNA. Libraries were individually barcoded and amplified by 6 PCR cycles before pooling. Sequencing was performed on multiple lanes of the Illumina HiSeq X platform in 150 bp paired end mode. Forty barcoded libraries were pooled in equimolar amounts per sequencing lane. Samples 16-237-021 and 16-284-032 were sequenced on the Ion S5 System. Single-end libraries were prepared with the Ion Plus Fragment Library Kit starting with 100 ng of DNA which was sheared to 400 bp using a Covaris M220 instrument. Libraries were amplified by 8 PCR cycles. Each library was sequenced on an individual Ion Torrent Chip 530 with a read length of 400 bp. Sequencing reads were quality-trimmed using Trimmomatic version 0.36[50] in the single-end mode (Ion Torrent reads) or paired-end mode (Illumina reads) using the following settings: LEADING:20 TRAILING:20 SLIDINGWINDOW:5:10 MINLEN:50. Adapter remnants were removed from the Illumina reads with the option: ILLUMINACLIP:TruSeq3-PE-2.fa:2:30:10. Ion Torrent adapters were removed in advance using Ion Torrent software.

Altogether, we sequenced 48 genomes from B.C. nurseries that we combined with 47 previously published genomes from Canada[20], the US[51] and Europe[20,52] for a total of 95 *P. ramorum* genomes. Data is available from NCBI bioprojects PRJNA791184, PRJNA427329, PRJNA559872, PRJNA177509, and PRJNA558041 (https://www.ncbi.nlm.nih.gov/bioproject). More information is provided in Supplementary Data 1.

**Variant calling and filtering.** Filtered reads from the genomes generated in this study and those obtained from other sources were mapped to the *P. ramorum* NA1 lineage reference genome PR-102_v3.1 using bwa mem version 0.7.17[53] with

default settings. Mapped reads were sorted using SAMtools version 1.9[54] and genome-wide average coverage was assessed using Qualimap version 2.2.1[55] with the bamqc option. Duplicated reads were annotated using the s (https://gatk.broadinstitute.org/hc/en-us). Variant calling was performed in two steps using Genome Analysis Toolkit (GATK;version 4.1.0.0). First, variants were called for each individual genome using GATK's HaplotypeCaller with the -ERC GVCF option and the ploidy was set to 2. Next, variant files were merged using GATK's CombineGVCFs function and then joint variant calling across all individuals was performed with GenotypeGVCFs. Raw variants were reduced to SNPs only and filtered using GATK's SelectVariants and VariantFiltration functionalities. Only SNPs satisfying the following criteria were retained: QUAL ≥ 500, QD ≥ 10, MQ ≥ 40, FS ≤ 40, SOR ≤ 3, and MQRankSum, BaseQRankSum, ReadPosRankSum ≥ −2 and ≤2. Read depth (DP) was restricted to ≥1000 (≈10% percentile) and ≤10000 (≈99% percentile) to further increase SNP quality and only biallelic SNPs were retained. Next, SNPs were filtered on the genotype level using VCFtools version 0.1.16 (https://vcftools.github.io/man_latest.html). Genotypes with a read depth <10 (−minDP 10) and a genotype quality <30 (−minGQ 30) were set to no-call, and variants with more than 5% missing genotypes were removed. Finally, invariant sites across all samples were removed. For some analyses (e.g., minimum spanning network, admixture) the SNP set was further restricted to variants without missing genotypes.

**Extraction of mitochondrial haplotypes**. Trimmed reads were mapped to the mitochondrial contig of the NA1 lineage reference genome PR-102_v3.1[56] (NCBI accession: NC_009384.1[57]) and variants called as described for the nuclear genome. Variants were restricted to SNPs only and the distribution of "variant confidence normalized by unfiltered depth of variant samples (QD)" was assessed to confirm the homozygous state across individuals. SNPs were filtered using GATK's SelectVariants and VariantFiltration functionalities with the following criteria: QUAL ≥ 1000, QD ≥ 20, MQ ≥ 59, FS ≤ 20, SOR ≤ 2 and only bi-allelic SNPs with no missing genotypes were accepted. VCFtools was then used to set genotypes with low support (read depth <10, i.e. −minDP 10 and genotype quality <40, i.e. −minGQ 40) to no-call. Sites with more than 5% missing genotypes and invariant SNPs across all individuals were removed.

**Population genomics analyses**. Population structuring and genetic connectivity of the unknown isolates with the four lineages of *P. ramorum* was evaluated using principal component analysis (PCA) with SNPrelate (version 1.28)[58]. Linkage disequilibrium-pruning was applied on the VCF dataset of 450,656 SNP loci to subsample 3,780 markers with reduced linkage ($R^2 < 0.20$). Ancestry estimation was conducted using Admixture among the *P. ramorum* lineages with the full SNP set with K = 4, the number of lineages found in North America and Europe. To assess reticulated relationships between *P. ramorum* individuals a neighbor-net phylogenetic network was reconstructed using SplitsTree version 4.12.8[59] with pairwise Nei's genetic distances with the R package StAMPP (version 1.6.3)[60].

**Genome-wide hybridization detection and gene flow simulations**. We used the function hybridization in the R package Adegenet (version 2.1.5) to simulate 10 F1 hybrids between NA2 and EU1 and 10 backcrosses of the hybrid to the EU1 or NA2 parent. We generated a pairwise distance matrix with the dist function (Euclidean distance) and obtained an NJ tree to visualize the position of the observed samples and simulation of hybrids and introgressants. We used the program HyDe (version 0.4.3)[61] to perform a genome-wide test of interspecific hybridization between *P. ramorum* lineages. HyDe considers a four-taxa network consisting of an outgroup and a triplet of ingroup taxa. The distribution of SNP site patterns in the triplet is used to infer a hybrid ingroup lineage that with a probability γ is sister to one lineage and with probability 1 – y is sister to the other lineage. Under the null hypothesis that admixture is absent, the expected value of y is zero. HyDe testing was performed with the 'run_hyde_mp.py' script of the HyDe package on all possible triplet combinations of putative parents-hybrid with NA1, NA2, EU1 and the 16-237-021 and 16-284-032 isolates (i.e., 12 triplets tested) while EU2 samples were used as outgroup.

**Haplotype phylogenetic network reconstruction**. Phased haplotypes were obtained by using short genome regions with physical phasing when two or more variants co-occur on the same sequencing read using WhatsHap (version 1.2)[62]. We tested the 15,265 genes annotated in the *P. ramorum* NA1 reference genome PR-102_v3.1 to identify genes containing at least 10 phased SNP loci in the two hybrid samples and samples from each *P. ramorum* lineage. For the conserved hypothetical protein, peroxisomal membrane protein and cell 12 A endoglucanase (Fig. 2a–c) the following genomes were used: 14-1270, 15-1093 (EU1), 04-0372 (NA1), 5-0954 (NA2) and P2111 (EU2). For the mitochondrial genomes (Fig. 2d), the following samples were used: 03-0110, 05-7036, 08-3469, BBA-26-02, CC1048, P2599 (EU1); 04-0372, MK548, Pr1556 (NA1), 04-25165, 05-18753, 13-0781 (NA2); P2111, P2460, P2461 (EU2). A FASTA format sequence alignment file containing two haplotype sequences for each *P. ramorum* samples (including the two hybrids) was then reconstructed as follows: the single nucleotide haplotypes predicted from a phased genotype were coded with their respective allele; the single

nucleotide haplotypes predicted from an unphased genotype were coded as missing data. Finally, sequence alignments were collapsed to unique haplotypes within each *P. ramorum* lineage. Maximum likelihood (ML) gene trees were inferred using RAxML[63] under the GTR model with a GAMMA parameter. Bootstrap support at nodes was determined by 1,000 iterations.

**Bayesian assignment test**. We designed an assignment procedure to test the most likely parents of the hybrid isolates. This procedure consisted in building a Bayesian naïve classifier model for *n* different classes (i.e., putative source populations) based on *k* SNP loci and obtaining the probability that an unknown *P. ramorum* sample belongs to each class. The assignment procedure was repeated 100 times with random samples of *k* = 1,000 loci. For each replicate, *n* = 14 putative source populations were considered: NA1 (29 diploid individuals), NA2 (29), EU1 (30) and EU2 (5). We simulated eight hybrid populations (of 50 individuals each) by randomly recombining SNPs from two parent lineages i.e. NA1 × NA2, NA1 × EU1, NA1 × EU2, NA2 × EU1, NA2 × EU2, EU1 × EU2. We also simulated populations backcrossed to one of the parents: (NA2 × EU1) × EU1 and (NA2 x EU1) × NA2. We simulated mutations with 50 random NA2 isolates for which 18.3% of the loci (randomly picked) were mutated by changing one of the alleles. This proportion of mutated loci was estimated by counting in 16_0237_0021 the number of loci for which at least one of the alleles could not have a NA2 origin, given the alleles found in the whole NA2 lineage for this locus (EstimateProportion.py script). The same procedure was done with 50 EU1 isolates with a proportion of mutated loci of 16.8%, to create the second "mutant" population. The naïve bayes classifier algorithm was implemented in a custom Python script using the Python package scikit-learn 0.24.1 and assuming Multinomial distribution of the data[64].

**Functional impact and phenotypic characterization**. To assess the potential functional impact of hybridization from genome predictions, we categorized the SNPs with regard to functional class and type using SnpEff (version 4.3) with default settings[65]. We used the *P. ramorum* reference genome (PR-102_v3.1) and the gff file to build the annotation database. Functional class consisted of missense (non-synonymous mutations); nonsense are mutations that cause stop codon loss or gain and frameshifts and silent mutations are synonymous. High impact mutations result in stop codon loss and gain and frameshifts; moderate impact mutations cause a codon to change the amino acid; low modifier mutations are in codons that produce the same amino acid; modifier mutations are in introns, upstream or downstream regions.

We measured growth of isolates of *P. ramorum* lineages and the EU1 × NA2 hybrids on carrot agar (CA) plates for four days at 20 °C; 7-mm plugs were transferred to fresh CA plates (3 replicates) and incubated in the dark at 20 °C. Isolates of the putative parental lineage NA2 (*n* = 11), EU1 (*n* = 11), and two hybrid isolates were used in the growth experiments. Host inoculations were performed in triplicate, using different rhododendron leaves. Rhododendron leaves were inoculated by placing a 7-mm plug from a 7-day-old colony of *P. ramorum* in the center of a detached leaf and measuring lesion development after 3, 4, 5, 6 and 10 days. The original data is available in Supplementary Data 2.

**Statistics and reproducibility**. For the HyDe analysis, we obtained average γ-values with standard deviations by subsampling the full dataset of 450,656 SNP loci 500 times for 10,000 random loci. The Z-statistic was used to test for significance of the y-values, and *p*-values were adjusted for multiple testing using Bonferroni adjustment (α = 0.05/12 = 0.0041). The growth and inoculation experiments (shown in Fig. 2) were done in triplicate using 11 different isolates of the pathogen for the NA2 and EU1 lineages and two hybrid isolates. The colony diameters were measured after 2, 3, 6, 7, and 9 days and mean daily radial growth rate was calculated using linear regression (lm(log(radius) ~ hours) in R. For the host inoculations, necrotic area was measured as the surface area with necrosis (mm$^2$) and averaged. One-way ANOVA with multiple comparisons using Tukey contrasts for multiple comparison of means were performed between lineages for growth and lesion size.

**Reporting summary**. Further information on research design is available in the Nature Research Reporting Summary linked to this article.

## Data availability
The data that support the findings of this study are included in the paper and all sequence data are deposited in NCBI GenBank as bioprojects, PRJNA791184, PRJNA427329, PRJNA559872, PRJNA177509, and PRJNA558041. Source data underlying Fig. 3 are presented in Supplementary Data 2.

## Code availability
The code used in this study is available available on GitHub (https://github.com/feaunico/SOD_hybrid/tree/v1.0) and Zenodo with https://doi.org/10.5281/zenodo.6465030.

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

## Acknowledgements
This work was supported by Genome Canada, Genome British Columbia, Genome Quebec, Natural Resources Canada, the Canadian Food Inspection Agency and FPInnovations through the Large Scale Applied Research Project (LSARP 10106. The authors acknowledge the help of Debbie Shearlaw, Miranda Newton, and Marie-Claude Gagnon from CFIA for technical assistance and Lucyna Kumor from Plant and Seed Pathology Laboratory at CFIA for the cultures.

## Author contributions
R.C.H., G.J.B., N.F., A.C., and R.H. conceived the project, obtained the samples, and performed the genome analyses. K.H., E.D., A.L.D., A.C., S.K., and E.G. performed the phenotyping. N.C.C and N.J.G. generated a reference genome that was used for mapping and analyses and provided annotations. R.C.H., N.F., R.H., and G.J.B. wrote the manuscript and the supplementary information with input from all other authors.

## Competing interests
The authors declare no competing interests.
