## [Peer Review File · Communications Biology]

Reviewers' comments:

Reviewer #1 (Remarks to the Author):

This work is based on an oomycete tree pathogen (*Phytophthora ramorum*), responsible for sudden oak death in western USA and sudden larch death in Europe. Strains of *P. ramorum* are clonal and thought to be reproductively isolated. The authors have sequenced isolates mainly from Europe and North America and provide good evidence for hybridisation between European and American lineages from this rhododendron nursery. The authors also detail hybrid phenotypes (growth rates and lesion size) as intermediate to their source lineages and discuss the implications of Transgressive Segregation. The work presented here is convincing and the manuscript is well written.

The authors stitch their finding (hybrids) into broader fields of the role of hybridisation in invasive pathogen lineages and the importance of biosurveillance methods developed to detect this. Genetic recombination in pathogens (I make no distinction between hybridisation, introgression and sex between divergent lineages) is one of the most important forces in pathogen evolution and invasion success. However, here I felt the authors' framed the text mostly within tree pathogens where they could have highlighted genetic reservoirs more broadly and potentially increased the impact of their work. On the side of biosurveillance too I thought the authors could have gone further to interpret their findings in terms that reader's, developing biosurveillance infrastructure, could directly interpret, I detail this further below. In short, there are no major criticisms of this work but, if they wish to, the authors could increase the broadness of its appeal in my opinion.

Minor comments

The authors should mention that this pathogen is an oomycete. I think it would be nice at least to say that these processes are occurring in other oomycetes as well as fungi and much broader (e.g. below):

Here the authors explore the result of hybridisation in a tree pathogen, an oomycete, but this work on hybridisation is equally valuable for researchers working on other pathogens. As is made clear in the title, the authors highlight biosurveillance as a means to detect these hybridisation events, often associated with colonisation of new hosts or regions. Both genetic hybridisation and biosurveillance are extremely interesting areas. However, I think that on the hybridisation side, the authors could increase the appeal of this ms to a broader audience. Examples of hybridisation and introgression, which drive speciation, facilitate host jumps and increase virulence have been identified in other tree pathogens (Brasier & Kirk 2010), plant fungi (Stukenbrock et al., 2012), and oomycetes (McMullan et al., 2015) but this work should be of interest to researchers much broader than plants as there is evidence that hybridisation was important for the generation of the human malaria pathogen (Galaway et al., 2019).

On the biosurveillance side, the authors only seem to mention that hybridisation is important for biosurveillance, they don't reconcile their data within a biosurveillance style approach. How do the data from this test system inform researchers developing biosurveillance strategies? The answers are all in the ms (heterozygosity and haplotype diversity etc. would be important metrics to determine to identify novel virulent invading lineages) but as it's currently written I feel that these two huge areas of biology are not integrated into the findings of the study.

Figure 2. I agree with the main conclusion that haplotype diversity within hybrids separates clearly into NA2 and EU1 lineages. But the green EU1 population looks to be quite large based on its genetic diversity in A and B. Moreover, in B EU1 appears to have representation on both of the main two lineages, as does NA1. The authors don't go on to discuss the reasons for this. If their hybrid sequences had aligned with one of these sequences would that have invalidated their finding, if not why not?

Are the two hybrid isolates they have identified from the same cross?

-Do their shared EU1 and NA2 haplotypes differ (i.e. is this a rare natural event or does it happen

all the time)? If so by how much on average do the haplotypes? Again if this is interesting how could this be incorporated into biosurveillance by informing on the likely rate of hybridisation.

In119 The authors measure growth rates and found they were intermediate. Transgressive segregation operates on the segregation of variation generated in an F2 progeny. The F1 (hybrids) contains whole chromosomes from each of the parental species and it is only the F2 generation that releases phenotypic diversity when these parental chromosomes are recombined together. Intermediate F1s is also a prediction of TS. Can they cross their hybrids? Do they produce vastly different offspring? In terms of statistics important to survey for biocontrol, a statistic that described breakdown of linkage in diverse haplotypes would be informative.

Fig3A. Are all three of these plates the same hybrid isolate? Would it not be worth comparing this isolate to the parental isolates?

In139 How did the authors ensure that their ROH were real in these TE regions and not an artifact of poor mapping in repetitive regions?

Points the authors may wish to ignore:
I would put the Latin name of their species in the title

I'm not suggesting the authors use these references I suggested above and I see already that they cite reviews by Stukenbrock and Braiser but I think the authors would increase their readership if they set their work more broadly within a context of the power of introgression/hybridisation to facilitate rapid adaptation and improve invasion potential. I add the references here for ease.

-Brasier CM, Kirk SA. Rapid emergence of hybrids between the two subspecies of *Ophiostoma novo-ulmi* with a high level of pathogenic fitness. *Plant Pathol.* 2010;59(1):186–99.

-Galaway F, Yu R, Constantinou A, Prugnolle F, Wright GJ. Resurrection of the ancestral RH5 invasion ligand provides a molecular explanation for the origin of *P. falciparum* malaria in humans. *PLoS Biol* [Internet]. 2019;17(10):e3000490. Available from:

<http://www.ncbi.nlm.nih.gov/pubmed/31613878>

-McMullan M, Gardiner A, Bailey K, Kemen E, Ward BJ, Cevik V, et al. Evidence for suppression of immunity as a driver for genomic introgressions and host range expansion in races of *Albugo candida*, a generalist parasite. *Elife.* 2015 Feb 27;4:1–24. Available from:

<http://elifesciences.org/lookup/doi/10.7554/eLife.04550>

-Stukenbrock EH, Christiansen FB, Hansen TT, Duteil JY, Schierup MH. Fusion of two divergent fungal individuals led to the recent emergence of a unique widespread pathogen species. *Proc Natl Acad Sci U S A* [Internet]. 2012 Jul 3;109(27):10954–9. Available from:

<http://www.pubmedcentral.nih.gov/articlerender.fcgi?artid=3390827&tool=pmcentrez&rendertype=abstract>

Reviewer #2 (Remarks to the Author):

Review of Hamelin et al., Genomic biosurveillance detects a sexual hybrid in the sudden oak death pathogen for *Communication Biology*.

The short study reported by Hamelin et al. is important as it describes the suspected sexual hybridization of two clonal lineages of *Phytophthora ramorum*. Sexual hybridization may generate novel isolates that present new challenges to control this important pathogen. While the study analyses a total of 95 isolates, the main focal point of the study are two isolates which represent putative hybrids. These two isolates are very similar, collected from a single nursery in British Columbia in 2016, and likely originated from a single hybridization event. Putative hybridization is proposed due to SNPs which are homozygous in the parental lineages being heterozygous in the putative sexual hybrids. Phylogenetic analysis of three genes supports the recovery of two distinct haplotypes that cluster with sequences from either parent. Polymorphisms called in the progeny are consistent with having segregated from the putative parents. Mitochondrial analysis suggests that only one mitotype is present, suggesting uni-parental inheritance. Presently, I think the inheritance of polymorphisms and mitochondrial analyses are convincing data for sexual

hybridization. However, the study may benefit from ruling out polyploidy or heterokaryosis. To this end, I believe the authors could further analyze the nuclear sequencing data they have in order to support their sexual hybridization hypothesis. I would suggest:

1. Investigating the raw reads to determine if both haplotypes are present at the same read-depths. If this is a sexual hybridization, then one haplotype from EU1 and one haplotype from NA2 should be present in each nucleus. If this is the case, SNPs specific to each haplotype should be present at approximately the same levels. Plotting a histogram of the allele balance frequencies of the 31,047 SNPs, homozygous in the putative parents, but heterozygous in the hybrids, for the higher coverage isolate may also provide insight as it should be centered at 0.5 in a sexual hybrid.

2. Investigate which haplotype has not been inherited by the sexual hybrids. Again, since one haplotype from EU1 and one from NA2 is present in the hybrids then some polymorphisms that differentiate these two clonal lineages should be missing from the hybrid isolates. Is this the case? Can the authors add the number of SNP alleles detected in the parents, not present in the putative progeny, perhaps to table S3? Demonstrating that one haplotype of each parental lineage was not present would be convincing evidence of sexual hybridization.

Given the potential significance of the sexual hybridization, did the authors attempt to generate single zoospore isolate? This would be beneficial to the study, as it would confirm that both EU1 and NA2 genotypes are present in the zoospores of hybrid isolates.

As a researcher interested in plant pathogenic oomycetes, I found the paper clear and enjoyable to read. I believe it will be a good addition to the current literature.

Additional comments.

The title leads "Genomic Biosurveillance...". What do the authors consider genomic biosurveillance? Is it widely applied to *P. ramorum*? Genomic biosurveillance is not mentioned in the introduction, only the last sentences of the abstract and discussion.

Should the abstract contain citations?

I found this sentence hard to interpret: "The pathogen comprises divergent clonal lineages¹⁵ that are reproductively isolated and confined to North America (NA1, NA2) or Europe (EU2), one that is broadly distributed in Europe and the West Coast of North America (EU1)^{16,17}, and additional lineages recently described in Asia¹⁸".

Line 80: The qPCR genotyping pattern is from a single gene, correct? If so, I think it is important to mention this as most readers will not be aware.

Line 82: "We sequenced the genomes of 95 *P. ramorum* isolates", but line 175, "We also retrieved previously sequenced genomes... for a total of 95 *P. ramorum* isolates". Therefore, this study did not sequence the genomes of 95 isolates?

Line 86: I recommend stating "principal component analysis of the 450,656 SNPs" to not confuse with the subset described in the previous sentence.

Line 102 and table S3: Do the authors mean meiotic recombination? Wouldn't sexual hybridization be a better term since the authors are predicting the genotypes of progeny (i.e. a Punnett square)? What do the authors mean by somatic recombination here? I am not sure how the authors can predict somatic recombination patterns given two parental genotypes? Should this instead be somatic hybridization/polyploidy?

Line 108: Do 103 mitochondrial SNPs differentiate EU1 mitotypes from NA2? As written, this is unclear to me. Were 100% of the sites concordant between NA2 and the hybrids?

Line 118: I recommend re-stating that the previous studies crossed EU1 x NA1.

Line 172: Please state how many isolates were collected from nurseries.

Line 173: Please provide a BioProject for newly generated sequences.

Line 176: Please provide specific BioProjects for each study? Currently the link is to the NCBI BioProject front page.

Mitochondrial analysis missing from the phylogenetic methods.

Figure 2 caption and results state neighbor joining trees. Methods state maximum likelihood. Which is correct? Are the numbers annotated on the branch bootstraps? If so, what is causing the low bootstrap confidence leading to the hybrid/NA2 polytomy?

Figure 2: Were alleles phased for other lineages? Why does EU1 have three alleles in tree A? Is NA2 homozygous in tree B? Are both EU1 and NA2 homozygous in tree C?

Figure 3: recommend authors highlight chlamydospores and sporangia in panels, mentioned on line 116.

Figure 3: How many measurements were made for panels D and E? Could a scatter plot be overlaid?

Dear Editor, please find below our answers to the comments of the Editor and the reviewers. Our answers are in bold italics.

Dear Professor Hamelin,

Your manuscript entitled "Genomic biosurveillance detects a sexual hybrid in the sudden oak death pathogen" has now been seen by 2 referees. You will see from their comments below that while they find your work of considerable interest, some important points are raised. We are interested in the possibility of publishing your study in Communications Biology, but would like to consider your response to these concerns in the form of a revised manuscript before we make a final decision on publication.

We therefore invite you to revise and resubmit your manuscript, taking into account the points raised. In particular, you would need to pay special attention to the following points for us to contact our referees again:

1. Reviewer 1's suggestion to broaden the framing of the manuscript by including additional references and examples, particularly in the introduction and discussion.

Examples and references were added to the introduction (first two paragraphs) and discussion (first paragraph) to highlight the impact of hybridization in a broad range of pathogens, including human pathogens.

2. Do also provide context regarding the term "genomic biosurveillance" as reviewer 2 notes this term is included in the title but it is not mentioned in the introduction and only in the last sentences of the abstract and discussion. Please expand accordingly. In addition, due to the focus on biosurveillance it would be helpful to reconcile your data within a biosurveillance style approach as suggested by reviewer 1.

The concept of genomic biosurveillance was expanded, by adding a sentence in the abstract, a paragraph at the end of the introduction and in the discussion. We also provide additional information, including the lineage variability as well as a new supplementary table (S10) with additional metadata.

3. Please consider the suggestions by reviewer 2 regarding ruling out polyploidy or heterokaryosis through additional analyses.

There are have several lines of evidence that rule out polyploidy. We added a figure S1 that shows that read proportion for the "minor" allele centers around

0.5, indicating that a large majority of these loci have only two alleles in equal proportion in the hybrids. Heterokaryosis or polyploidy would generate heterogeneous read coverage, which we did not observe. Another observation that does not support heterokaryosis caused by fusion of the EU1 and NA2 lineages is the absence of the EU1 mitochondrial lineage in the hybrids. Please, see answer to comment #1 of Reviewer 2 for more extensive explanations.

4. Please consider also the suggestion regarding generation of single zoospores to confirm that EU1 and NA2 are present in the hybrids.

Although generating single zoospore isolates of the hybrid would be worthwhile, we are uncertain what this would add to our discovery of the hybrids. The isolates were obtained by single hyphal tip cultures and the genomic pattern that we observed is best explained by hybridization. We rejected the hypothesis of heterokaryosis via hyphal fusion by comparing observed and expected allele frequency and by showing the absence of EU1 mitotype in the hybrid. Since the zoospores would be produced via mitosis from the mycelium in the cultures, we would expect to obtain the same patterns as in the cultures. We expand on these explanations below.

5. Please do also carefully consider all the minor points highlighted by the reviewers.

The manuscript was carefully edited according to the reviewer's comment and the comments from the Editor. In addition, we moved the methods from the supplementary file to the main document to make it easier for the reader to refer to the methods. We have now also consolidated and expanded the Discussion.

Please highlight all changes in the manuscript text file.

We used track changes to make the evaluation of the changes easier.

We are committed to providing a fair and constructive peer-review process. Do not hesitate to contact us if you wish to discuss the revision in more detail or if there are specific requests from the reviewers that you believe are technically impossible or unlikely to yield a meaningful outcome.

At the same time, we ask that you ensure your manuscript complies with our editorial policies. Please see our revision file checklist for guidance on formatting the manuscript and complying with our policies. You will also find guidelines for replying to the referees'

comments. You may also wish to review our formatting guidelines for final submissions here.

Please use the following link to submit your revised manuscript, point-by-point response to the referees' comments (which should be in a separate document to the cover letter) and any additional files:

[https://mts-commsbio.nature.com/cgi-](https://mts-commsbio.nature.com/cgi-bin/main.plex?el=A1Cx7Drm5A2vOA1I5A9ftd0A7IrKfFa8334jck36fqgZ)

[bin/main.plex?el=A1Cx7Drm5A2vOA1I5A9ftd0A7IrKfFa8334jck36fqgZ](https://mts-commsbio.nature.com/cgi-bin/main.plex?el=A1Cx7Drm5A2vOA1I5A9ftd0A7IrKfFa8334jck36fqgZ)

When submitting the revised version of your manuscript, please pay close attention to our Digital Image Integrity Guidelines.

We would expect revisions of this nature to take around three months, but appreciate that every situation is unique. We look forward to receiving your revised manuscript when it is ready, and will not enforce a hard deadline on this revision.

Please do not hesitate to contact me if you have any questions or would like to discuss these revisions further. We look forward to seeing the revised manuscript and thank you for the opportunity to review your work.

Best regards,

Diane Saunders, PhD
Editorial Board Member
Communications Biology
orcid.org/0000-0003-2847-5721

Reviewers' comments:

Reviewer #1 (Remarks to the Author):

This work is based on an oomycete tree pathogen (*Phytophthora ramorum*), responsible for sudden oak death in western USA and sudden larch death in Europe. Strains of *P. ramorum* are clonal and thought to be reproductively isolated. The authors have sequenced isolates mainly from Europe and North America and provide good evidence

for hybridisation between European and American lineages from this rhododendron nursery. The authors also detail hybrid phenotypes (growth rates and lesion size) as intermediate to their source lineages and discuss the implications of Transgressive Segregation. The work presented here is convincing and the manuscript is well written.

The authors stitch their finding (hybrids) into broader fields of the role of hybridisation in invasive pathogen lineages and the importance of biosurveillance methods developed to detect this. Genetic recombination in pathogens (I make no distinction between hybridisation, introgression and sex between divergent lineages) is one of the most important forces in pathogen evolution and invasion success. However, here I felt the authors' framed the text mostly within tree pathogens where they could have highlighted genetic reservoirs more broadly and potentially increased the impact of their work.

We have broadened this topic by adding a few sentences in the second paragraph of the Introduction about genetic changes generated by hybridization and providing examples with human parasites and plants.

On the side of biosurveillance too I thought the authors could have gone further to interpret their findings in terms that reader's, developing biosurveillance infrastructure, could directly interpret, I detail this further below. In short, there are no major criticisms of this work but, if they wish to, the authors could increase the broadness of its appeal in my opinion.

The concept of genomic biosurveillance was expanded, by adding a sentence in the abstract, a paragraph at the end of the Introduction and in the Discussion. We also added data on diversity in the lineages and additional metadata.

Minor comments

The authors should mention that this pathogen is an oomycete.

Done.

I think it would be nice at least to say that these processes are occurring in other oomycetes as well as fungi and much broader (e.g. below)

Done. We added references to hybridization in fungi, plants and protozoa.

Here the authors explore the result of hybridisation in a tree pathogen, an oomycete, but this work on hybridisation is equally valuable for researchers working on other pathogens. As is made clear in the title, the authors highlight biosurveillance as a means to detect these hybridisation events, often associated with colonisation of new hosts or regions. Both genetic hybridisation and biosurveillance are extremely interesting areas. However, I think that on the hybridisation side, the authors could increase the appeal of this ms to a broader audience. Examples of hybridisation and introgression, which drive speciation, facilitate host jumps and increase virulence have been identified in other tree pathogens (Brasier & Kirk 2010), plant fungi (Stukenbrock et al., 2012), and oomycetes (McMullan et al., 2015) but this work should be of interest to researchers much broader than plants as there is evidence that hybridisation was important for the generation of the human malaria pathogen (Galaway et al., 2019).

Done, we added some of those references.

On the biosurveillance side, the authors only seem to mention that hybridisation is important for biosurveillance, they don't reconcile their data within a biosurveillance style approach. How do the data from this test system inform researchers developing biosurveillance strategies? The answers are all in the ms (heterozygosity and haplotype diversity etc. would be important metrics to determine to identify novel virulent invading lineages) but as it's currently written I feel that these two huge areas of biology are not integrated into the findings of the study.

The idea of using a biosurveillance approach is to document, via genome sequencing of pathogen samples surveyed across spatial and temporal scales, genomic patterns that could identify novel variants, recombinants or hybrids. We added sections in the introduction and discussion to contextualize and emphasize this concept. Heterozygosity is also an important measure that could separate clonal from recombining populations. We agree with the reviewer that some of the measures, such as heterozygosity, are important to report. We are now reporting observed heterozygosity in the hybrids and the other lineages and present these in the new supplementary Table S4. However, we do not report haplotype diversity because we do not have phased haplotypes (except for the few genes for which we were able to obtain them, presented in Fig. 2).

Figure 2. I agree with the main conclusion that haplotype diversity within hybrids separates clearly into NA2 and EU1 lineages. But the green EU1 population looks to be quite large based on its genetic diversity in A and B. Moreover, in B EU1 appears to have representation on both of the main two lineages, as does NA1. The authors don't

go on to discuss the reasons for this. If their hybrid sequences had aligned with one of these sequences would that have invalidated their finding, if not why not?

Indeed, the reviewer is correct that EU1 seems to be the most diverse lineage, based on our results here, with the highest heterozygosity (new Table S4), but also based on our previous publication on intralinear diversity in *P. ramorum* (Dale et al. 2019 <https://doi.org/10.1128/mBio.02452-18>). Diversity was observed in all lineages of *P. ramorum* that were caused by runs of homozygosity, cnv and single nucleotide polymorphisms. This points to an evolutionary process that allows the pathogen to accumulate genomic changes even in the absence of sexual reproduction. Thus, it is not unexpected to have more than two alleles at a specific gene (e.g. three haplotypes for EU1 in Fig. 2A). In Fig. 2B, one of the two haplotypes obtained for the hybrid samples (light blue) matches an NA2 haplotype and the other one matches a EU1 haplotype; this is exactly the pattern to be expected for a F1 hybrid. However, one of the alleles at the locus (on the top node) appears to be conserved in NA1, NA2 and EU1. Yet, only the hybrid haplotype completely matches the NA2 haplotype and the EU1 and NA1 haplotypes are still quite well differentiated (branch with ~0.002 substitution/site which correspond to 2 or 3 mutations on a sequence alignment of 1260nt.).

Are the two hybrid isolates they have identified from the same cross?

-Do their shared EU1 and NA2 haplotypes differ (i.e. is this a rare natural event or does it happen all the time)?

Our results point to a single recombination event followed by asexual propagation. The two hybrid samples that we identified have near-identical genome sequences (with approximately 0.16% divergence between the two isolates, most of which is likely attributable to small ROH and/or sequencing errors), and the phased haplotypes between the two hybrids have identical sequences. The sites that are fixed homozygous in parental lineages but heterozygous in the hybrids (first line in Table S3) are at the same positions in the two hybrid samples. We have now sequenced additional samples (not shown here), as part of a larger biosurveillance study, and found no other instance of hybridization, confirming that this was a rare and isolated event.

If so by how much on average do the haplotypes? Again if this is interesting how could this be incorporated into biosurveillance by informing on the likely rate of hybridisation.

Given the level of divergence between the lineages of *P. ramorum*, hybridization between lineages would generate a clear pattern in a genomic biosurveillance program. Certainly, the F1 hybrids would be clearly identifiable. Even introgression to one of the parents would generate a distinct signature, as can be seen in the simulations in Fig. S2. The advantage of a genomic biosurveillance program would be to provide early identification of such hybridization or introgression events. Even a targeted resequencing approach (which we are currently developing) can detect those hybrids by sequencing ~200 amplicons. Our results show that the rate of hybridization is very low, at least in nurseries. Further sampling in the Oregon forests is underway to determine if hybridization is more frequent in natural forests where all three *P. ramorum* lineages occur.

In119 The authors measure growth rates and found they were intermediate. Transgressive segregation operates on the segregation of variation generated in an F2 progeny. The F1 (hybrids) contains whole chromosomes from each of the parental species and it is only the F2 generation that releases phenotypic diversity when these parental chromosomes are recombined together. Intermediate F1s is also a prediction of TS. Can they cross their hybrids? Do they produce vastly different offspring? In terms of statistics important to survey for biocontrol, a statistic that described breakdown of linkage in diverse haplotypes would be informative.

These are excellent comments and suggestions. However, experimental sexual crosses in *P. ramorum* have been very difficult to perform and mostly yield non-viable progeny (<https://doi.org/10.1016/j.fgb.2011.01.008>). Further elucidation of potential future introgressants, as suggested by the reviewer, is a research area that is being currently pursued. However, it goes beyond the scope of the current study, which is to report the discovery interlineage F1 hybrids and characterize their genomes and phenotypes.

Fig3A. Are all three of these plates the same hybrid isolate? Would it not be worth comparing this isolate to the parental isolates?

The figure represents three cultures of one of the hybrids, but the hybrids were also grown and compared to the other lineages, including the putative parental lineages. In Fig. 3D, we are comparing the growth and virulence (aggressiveness) of the hybrids and the two putative parental lineages (NA2 and EU1) as well as the third lineage (NA1) that was not involved in the hybridization event.

In139 How did the authors ensure that their ROH were real in these TE regions and not an artifact of poor mapping in repetitive regions?

SNP loci in these regions were filtered with stringent parameters (as for the entire genome) as described in the M&M paragraph entitled Variant calling and filtering. New ROH, generated by mitotic recombination IN the hybrid, where then identified by looking for stretches of homozygous regions as detailed in Dale et al. (2019; <https://doi.org/10.1128/mBio.02452-18>) and identifying those in which more than 50% of the SNPs were either fixed homozygous for different alleles in the parental lineages and homozygous in the hybrid or fixed homozygous for one allele in one of the parental lineages, homozygous for the alternate allele in the hybrid and fixed heterozygote in the other parental lineage. These patterns are unexpected in a first generation (hybrid generated by sexual recombination and indicate the possibility of mitotic recombination happening during asexual propagation of the hybrid. Table S9 has been updated with these results.

Points the authors may wish to ignore:

I would put the Latin name of their species in the title

We prefer to have the latin name in the abstract, not the title, to broaden the appeal.

I'm not suggesting the authors use these references I suggested above and I see already that they cite reviews by Stukenbrock and Braiser but I think the authors would increase their readership if they set their work more broadly within a context of the power of introgression/hybridisation to facilitate rapid adaptation and improve invasion potential. I add the references here for ease.

-Brasier CM, Kirk SA. Rapid emergence of hybrids between the two subspecies of *Ophiostoma novo-ulmi* with a high level of pathogenic fitness. *Plant Pathol.* 2010;59(1):186–99.

-Galaway F, Yu R, Constantinou A, Prugnolle F, Wright GJ. Resurrection of the ancestral RH5 invasion ligand provides a molecular explanation for the origin of *P. falciparum* malaria in humans. *PLoS Biol* [Internet]. 2019;17(10):e3000490. Available from: <http://www.ncbi.nlm.nih.gov/pubmed/31613878>

-McMullan M, Gardiner A, Bailey K, Kemen E, Ward BJ, Cevik V, et al. Evidence for suppression of immunity as a driver for genomic introgressions and host range expansion in races of *Albugo candida*, a generalist parasite. *Elife.* 2015 Feb 27;4:1–24. Available from: <http://elifesciences.org/lookup/doi/10.7554/eLife.04550>

-Stukenbrock EH, Christiansen FB, Hansen TT, Dutheil JY, Schierup MH. Fusion of two divergent fungal individuals led to the recent emergence of a unique widespread pathogen species. Proc Natl Acad Sci U S A [Internet]. 2012 Jul 3;109(27):10954–9.

Available from:

<http://www.pubmedcentral.nih.gov/articlerender.fcgi?artid=3390827&tool=pmcentrez&rendertype=abstract>

We incorporated some of these references

Reviewer #2 (Remarks to the Author):

Review of Hamelin et al., Genomic biosurveillance detects a sexual hybrid in the sudden oak death pathogen for Communication Biology.

The short study reported by Hamelin et al. is important as it describes the suspected sexual hybridization of two clonal lineages of *Phytophthora ramorum*. Sexual hybridization may generate novel isolates that present new challenges to control this important pathogen. While the study analyses a total of 95 isolates, the main focal point of the study are two isolates which represent putative hybrids. These two isolates are very similar, collected from a single nursery in British Columbia in 2016, and likely originated from a single hybridization event. Putative hybridization is proposed due to SNPs which are homozygous in the parental lineages being heterozygous in the putative sexual hybrids. Phylogenetic analysis of three genes supports the recovery of two distinct haplotypes that cluster with sequences from either parent. Polymorphisms called in the progeny are consistent with having segregated from the putative parents. Mitochondrial analysis suggests that only one mitotype is present, suggesting uni-parental inheritance. Presently, I think the inheritance of polymorphisms and mitochondrial analyses are convincing data for sexual hybridization. However, the study may benefit from ruling out polyploidy or heterokaryosis. To this end, I believe the authors could further analyze the nuclear sequencing data they have in order to support their sexual hybridization hypothesis. I would suggest:

1. Investigating the raw reads to determine if both haplotypes are present at the same read-depths. If this is a sexual hybridization, then one haplotype from EU1 and one haplotype from NA2 should be present in each nucleus. If this is the case, SNPs specific to each haplotype should be present at approximately the same levels. Plotting a histogram of the allele balance frequencies of the 31,047 SNPs, homozygous in the putative parents, but heterozygous in the hybrids, for the higher coverage isolate may also provide insight as it should be centered at 0.5 in a sexual hybrid.

Done. We added a new supplementary figure (Fig. S1) showing the distribution of sequencing reads for the “minor” allele at heterozygous loci for the two hybrid samples. This distribution is centered at 0.5 indicating that these loci have only two alleles and that they are in equal proportion. We also performed a chi-square goodness of fit test at each heterozygous locus for the allele ratio 1:1 expected under the hypothesis of homoploid hybridization. According to this test, >90% of the loci tested were consistent with this expectation.

2. Investigate which haplotype has not been inherited by the sexual hybrids. Again, since one haplotype from EU1 and one from NA2 is present in the hybrids then some polymorphisms that differentiate these two clonal lineages should be missing from the hybrid isolates. Is this the case? Can the authors add the number of SNP alleles detected in the parents, not present in the putative progeny, perhaps to table S3? Demonstrating that one haplotype of each parental lineage was not present would be convincing evidence of sexual hybridization.

We believe that Fig. 2 shows this very clearly, with the hybrids possessing one allele (haplotype) from each putative parent, but missing the alternative allele(s). Also, Table S3 shows all the possible genotypic combinations in the hybrids and the parents and the observed values are remarkably similar to those expected.

Given the potential significance of the sexual hybridization, did the authors attempt to generate single zoospore isolate? This would be beneficial to the study, as it would confirm that both EU1 and NA2 genotypes are present in the zoospores of hybrid isolates.

Both isolates of the pathogen were generated from single hyphal tips. We did obtain sporangia from the cultures (Fig. 3B), but given that zoospores are asexual diploid spores, we are not sure what we would learn by genotyping the zoospores. Clearly, they are derived from the mycelium which we showed is an NA2 x EU1 hybrid, so the zoospores are expected to have identical genotypes as the mycelium they are derived from. The genomic profiles that we obtained can be best explained by hybridization, not heterokaryosis, which would generate different profiles from those we observed. Recreating the hybrid in vitro by obtaining crosses between NA2 and EU1 or conducting backcrosses between the hybrids and the parental lineages would be more interesting, but as indicated above, is beyond the scope of this study.

As a researcher interested in plant pathogenic oomycetes, I found the paper clear and enjoyable to read. I believe it will be a good addition to the current literature.

Additional comments.

The title leads “Genomic Biosurveillance...”. What do the authors consider genomic biosurveillance? Is it widely applied to *P. ramorum*? Genomic biosurveillance is not mentioned in the introduction, only the last sentences of the abstract and discussion.

As mentioned above, we are now clarifying this aspect.

Should the abstract contain citations?

We removed the citations in the abstract.

I found this sentence hard to interpret: “The pathogen comprises divergent clonal lineages¹⁵ that are reproductively isolated and confined to North America (NA1, NA2) or Europe (EU2), one that is broadly distributed in Europe and the West Coast of North America (EU1)^{16,17}, and additional lineages recently described in Asia¹⁸”.

We fixed this sentence.

Line 80: The qPCR genotyping pattern is from a single gene, correct? If so, I think it is important to mention this as most readers will not be aware.

A single gene, but 4 SNPs are genotyped for the lineage identification. This is now clarified in Table S1 and in the M&M.

Line 82: “We sequenced the genomes of 95 *P. ramorum* isolates”, but line 175, “We also retrieved previously sequenced genomes... for a total of 95 *P. ramorum* isolates”. Therefore, this study did not sequence the genomes of 95 isolates?

Right!! Fixed. We provide now a new table S10 that describes the metadata as well as the Bioproject numbers to allow the reader to obtain this information easily.

Line 86: I recommend stating “principal component analysis of the 450,656 SNPs” to not confuse with the subset described in the previous sentence.

Done

Line 102 and table S3: Do the authors mean meiotic recombination? Wouldn't sexual hybridization be a better term since the authors are predicting the genotypes of progeny

(i.e. a Punnett square)? What do the authors mean by somatic recombination here? I am not sure how the authors can predict somatic recombination patterns given two parental genotypes? Should this instead be somatic hybridization/polyploidy?

We are now clarifying this point by contrasting sexual recombination with heterokaryosis generated by hyphal fusion.

Line 108: Do 103 mitochondrial SNPs differentiate EU1 mitotypes from NA2? As written, this is unclear to me. Were 100% of the sites concordant between NA2 and the hybrids?

Done. We are now providing details on the mitochondrial genotypes, the mitotypes, in the Results section.

Line 118: I recommend re-stating that the previous studies crossed EU1 x NA1.

Done

Line 172: Please state how many isolates were collected from nurseries.

Done. This is now provided in the new Table S10.

Line 173: Please provide a BioProject for newly generated sequences.

Done. This is now provided in the new Table S10.

Line 176: Please provide specific BioProjects for each study? Currently the link is to the NCBI BioProject front page.

The information is now provided in Table S10.

Mitochondrial analysis missing from the phylogenetic methods.

Done. See “Haplotype phylogenetic network reconstruction” paragraph

Figure 2 caption and results state neighbor joining trees. Methods state maximum likelihood. Which is correct? Are the numbers annotated on the branch bootstraps? If so, what is causing the low bootstrap confidence leading to the hybrid/NA2 polytomy?

Figure 2: Were alleles phased for other lineages? Why does EU1 have three alleles in tree A? Is NA2 homozygous in tree B? Are both EU1 and NA2 homozygous in tree C?

Alleles were phased for all lineages and the hybrids at these selected loci. As mentioned earlier, it is not unexpected to have more than two alleles at a specific gene (with a [few] point mutation[s] between the different alleles), given the level of intralinear diversity that we are showing here and elsewhere (Dale et al. 2019). Yes, in tree B, NA2 is homozygous. In Fig. 2B, one of the two haplotypes obtained for the hybrid samples (light blue) matches an NA2 haplotype and the other one matches a EU1 haplotype; this is exactly the pattern to be expected for a F1 hybrid. However, one of the alleles at the locus (on the top node) appears to be conserved in NA1, NA2 and EU1. Yet, only the hybrid haplotype completely matches the NA2 haplotype and the EU1 and NA1 haplotypes still quite well differentiated (branch with ~0.002 substitution/site which correspond to 2 or 3 mutations on an alignment of 1260nt.).

Figure 3: recommend authors highlight chlamydospores and sporangia in panels, mentioned on line 116.

Fig. 3B shows only sporangia, which are abundant and do not require highlighting.

Figure 3: How many measurements were made for panels D and E? Could a scatter plot be overlaid?

As mentioned in the Materials and Methods: “The colony diameters were measured after 2, 3, 6, 7 and 9 days and mean daily radial growth rate was calculated using linear regression ($\ln(\log(\text{radius})) \sim \text{hours}$) in R.” For the leaf inoculations: “*Rhododendron* leaves were inoculated by placing a 7-mm plug from a 7-day-old colony of *P. ramorum* in the center of a detached leaf and measuring lesion development after 3, 4, 5, 6 and 10 days.”

REVIEWERS' COMMENTS:

Reviewer #1 (Remarks to the Author):

Further to comments from a previous review, all comments have been addressed and this work is acceptable in its current state.

Reviewer #3 (Remarks to the Author):

I believe the authors have improved their manuscript "Genomic biosurveillance detects a sexual hybrid in the sudden oak death pathogen" after review. I still find this an enjoyable and thought-provoking read.

In particular, I am happy to accept the evidence of sexual recombination from table S3 now that they have clarified the headers. I would recommend this table be promoted to the full manuscript as, in my opinion, it provides stronger evidence for sexual hybridization than Figure 2. I leave this to the discretion of the authors and editor.

For Figure 2, I believe it would be beneficial to readers of this paper to state which three isolates of NA1, EU1, NA2, and EU2 were surveyed. Otherwise, the work is not reproducible. In addition, it may be beneficial to state in the figure legend that either three isolates were used for each lineage or that "it is not unexpected to have more than two alleles at a specific gene" for EU1 isolates in tree A.

In addition, I do not believe it would be unreasonable to provide a scatter plot for Figure 3, as box plots only provide a summary of the data. Alternatively, the authors could provide a supplemental data file detailing the measurements.

Otherwise, I am satisfied with the response to my comments.

REVIEWERS' COMMENTS:

Reviewer #1 (Remarks to the Author):

Further to comments from a previous review, all comments have been addressed and this work is acceptable in its current state.

⇒ Thank you!

Reviewer #3 (Remarks to the Author):

I believe the authors have improved their manuscript “Genomic biosurveillance detects a sexual hybrid in the sudden oak death pathogen” after review. I still find this an enjoyable and thought-provoking read.

⇒ Thank you!

In particular, I am happy to accept the evidence of sexual recombination from table S3 now that they have clarified the headers. I would recommend this table be promoted to the full manuscript as, in my opinion, it provides stronger evidence for sexual hybridization than Figure 2. I leave this to the discretion of the authors and editor.

⇒ Thank you for the suggestion. We still think that this table is better suited to the supplementary material because of its complexity and Fig 2 provides a more visual representation. The readers can still access the table in the supplementary material.

For Figure 2, I believe it would be beneficial to readers of this paper to state which three isolates of NA1, EU1, NA2, and EU2 were surveyed. Otherwise, the work is not reproducible. In addition, it may be beneficial to state in the figure legend that either three isolates were used for each lineage or that “it is not unexpected to have more than two alleles at a specific gene” for EU1 isolates in tree A.

⇒ We are now providing the list of all isolates used for Figure 2 in the Materials and Methods.

In addition, I do not believe it would be unreasonable to provide a scatter plot for Figure 3, as box plots only provide a summary of the data. Alternatively, the authors could provide a supplemental data file detailing the measurements.

⇒ We are now provide a supplementary data 2 that gives the original data.

Otherwise, I am satisfied with the response to my comments.

⇒ Thank you!